# Histone Modifications in NAFLD: Mechanisms and Potential Therapy

**DOI:** 10.3390/ijms241914653

**Published:** 2023-09-27

**Authors:** Yulei Shi, Wei Qi

**Affiliations:** Gene Editing Center, School of Life Science and Technology, ShanghaiTech University, Shanghai 201210, China; shiyl@shanghaitech.edu.cn

**Keywords:** NAFLD, epigenetics, histone modifications, mechanism, therapeutic implications

## Abstract

Nonalcoholic fatty liver disease (NAFLD) is a progressive condition that encompasses a spectrum of liver disorders, beginning with the simple steatosis, progressing to nonalcoholic steatohepatitis (NASH), and possibly leading to more severe diseases, including liver cirrhosis and hepatocellular carcinoma (HCC). In recent years, the prevalence of NAFLD has increased due to a shift towards energy-dense dietary patterns and a sedentary lifestyle. NAFLD is also strongly associated with metabolic disorders such as obesity and hyperlipidemia. The progression of NAFLD could be influenced by a variety of factors, such as diet, genetic factors, and even epigenetic factors. In contrast to genetic factors, epigenetic factors, including histone modifications, exhibit dynamic and reversible features. Therefore, the epigenetic regulation of the initiation and progression of NAFLD is one of the directions under intensive investigation in terms of pathogenic mechanisms and possible therapeutic interventions. This review aims to discuss the possible mechanisms and the crucial role of histone modifications in the framework of epigenetic regulation in NAFLD, which may provide potential therapeutic targets and a scientific basis for the treatment of NAFLD.

## 1. Introduction

### 1.1. Pathogenesis of Nonalcoholic Fatty Liver Disease

Nonalcoholic fatty liver disease (NAFLD) is an increasingly common condition [1], which is strongly associated with metabolic diseases such as obesity, type 2 diabetes mellitus (T2DM), hyperlipidemia, and atherosclerosis [2,3]. NAFLD patients with metabolic syndrome or T2DM typically have an increased risk of death [2]. The prevalence of NAFLD ranges from 13.5% in Africa to 31.8% in the Middle East, and NAFLD is now the leading cause of chronic liver disease worldwide. In Western countries, the prevalence of NAFLD is 20–40% in adults, and 10–30% of the NAFLD patients eventually develop nonalcoholic steatohepatitis (NASH) [4]. Researchers have indicated that NASH patients who progress to the stage of cirrhosis have an increased risk of hepatocellular carcinoma (HCC) [5]. However, there is no approved pharmacotherapy for the treatment of NASH or liver cirrhosis. Therefore, NAFLD is a prevalent disease with a high social burden and unmet medical need.

From simple steatosis to NASH, cirrhosis, and hepatocellular cancer, NAFLD is a progressive disease [6] (Figure 1). Simple steatosis, or nonalcoholic fatty liver (NAFL), is characterized by the accumulation of triglycerides (TG) in hepatocytes. Hepatic steatosis can occur due to various factors. Excess dietary fat is certainly one. In addition, dietary carbohydrates raise blood glucose and insulin levels, leading to the activation of carbohydrate responsive element-binding protein (ChREBP) and sterol regulatory element-binding protein-1c (SREBP-1c) in the liver. These are two major transcription factors upregulating genes in fatty acid synthesis, thereby promoting de novo lipogenesis (DNL) [7,8,9], which produces free fatty acids (FFAs) from acetyl-coenzyme A (CoA) and leads to the accumulation of lipid droplets in hepatocytes upon activation. Furthermore, insulin resistance (IR), which is a common condition in patients with obesity or T2DM, could decrease the capacity of adipose tissue to store lipids, thereby increasing the amount of FFAs in the blood and, eventually, in the liver. In addition to producing energy through oxidation and storage in hepatocytes as TG, hepatic FFAs can be coupled to apolipoproteins and secreted in very-low-density lipoproteins (VLDL). Hepatic steatosis occurs when the amount of hepatocyte triglyceride production increases more than the amount of VLDL triglyceride release [10]. Therefore, an imbalance in lipid homeostasis causes the accumulation of lipids in the liver.

Unlike simple steatosis, NASH is a state in which inflammation and fibrosis occur in addition to the lipid accumulation in the liver. Therefore, NASH is characterized by the accumulation of excessive FFAs, increased oxidative/endoplasmic reticulum (ER)stress, fibrosis, and inflammation in the liver [11,12]. It is defined clinically by the presence of steatosis, hepatocellular ballooning, and lobular inflammation, accompanied by variable degrees of fibrosis upon a liver biopsy [13]. When there is an increased uptake/synthesis or hampered removal of fatty acids, they can be used as substrates for the production of lipotoxic species that induce cellular stress, hepatocyte apoptosis, and liver injury [14]. Indeed, the occurrence of NASH is associated with a number of cellular stresses, such as endoplasmic reticulum (ER) stress, mitochondrial damage, and oxidative stress, which is related to the production of reactive oxygen species (ROS) (Figure 1). It is noteworthy that NASH is progressive, and more severe forms may occur, such as liver cirrhosis and HCC [11]. HCC is the fifth-most prevalent form of cancer and the third leading cause of cancer-related death [15], highlighting the importance of preventing the progression of hepatic steatosis to HCC.

### 1.2. Factors Influencing NAFLD

NAFLD is a complex disease resulting from multiple aspects of lifestyle, diet, genetic variants, and epigenetic factors (Figure 2). The lack of physical exercise has been associated with increased body weight, insulin resistance, and an increased risk of metabolic syndrome and NAFLD [16]. Unhealthy diets may lead to overnutrition characterized by the excessive intake of fats or fructose, which are metabolized in the liver through DNL and converted to TGs. Consequently, this metabolic pathway contributes to the accumulation of fat in the liver. Insulin resistance has long been acknowledged as a crucial factor in the development of NAFLD [17]. There is a growing body of evidence supporting that the microbiome also plays a critical role in NAFLD [18,19,20]. The gut–liver axis has been connected to a number of disorders associated with obesity, including NAFLD, and these two organs are interdependent at many levels [21].

In addition to external or environmental cues, genetic variants significantly influence the progression of NAFLD [22]. One solid example is the genetic variant in the *PNPLA3* gene (patatin-like phospholipase domain-containing 3, rs738409), a substitution of cytosine with guanine resulting in a change of codon 148 from isoleucine to methionine. PNPLA3 is involved in the lipolysis of lipid droplets in hepatocytes. The I148M variant escapes proteasome degradation, accumulates on lipid droplets, and blocks the function of adipose triglyceride lipase and lipolysis [23]. PNPLA3-I148M has exhibited a robust correlation with increased hepatic TG accumulation and hepatic inflammation in human genome-wide association studies [24]. PNPLA3-I148M expression in liver resulted in an increase in the liver triglyceride content in a transgenic mouse model [25]. Thus, it is an important genetic risk factor and a valid therapeutic target for NAFLD [26]. Similarly, a few other genetic risk factors have been identified, such as a splice variant (rs72613567 T>A) in *HSD17B13* and an E to K substitution (rs58542926 C>T) in *TM6SF2*, which have been well summarized recently [27,28].

Due to the fact that epigenetic alterations can be dynamic and reversible, epigenetic mechanisms have been widely implicated in the initiation and progression of NAFLD, even in collaboration with other environmental cues (Figure 2). Epigenetic inheritance studies the changes in heritable gene expression or cellular expression caused by specific mechanisms without altering the DNA sequence [29]. Alterations in DNA methylation, histone variants and modifications, chromatin remodeling, and non-coding RNA-based mechanisms may all result in epigenetic changes. Several epigenetic mechanisms are of significant importance in the NAFLD spectrum of diseases [30]. For example, the cytosine methylation (5mC) of mitochondrial DNA (mtDNA) had been found in the liver biopsies of patients with NAFLD, and patients with NASH had higher levels of DNA methyltransferase 1 [31]. In another study, AAV-miR-20b administration induced hepatic steatosis and reduced FA oxidation in HFD-fed mice, possibly by decreasing the level of *PPARα* [32]. More broadly, histone modification changes may lead to the dysregulation of multiple biological processes associated with NAFLD, such as hepatic lipid accumulation, ER stress, oxidative stress, mitochondrial damage, and inflammation, which may be used as a single mechanism or work in synergy with the environmental factors on the development of NAFLD [22]. Due to length limitations, in this review, we focus on the regulatory mechanism of histone modifications, their pathological implications, and the potential therapeutic applications in the treatment of NAFLD.

## 2. Regulation of NAFLD by Histone Modifications

The histone core is an octamer, comprising two H2A, H2B, H3, and H4 molecules. The histone core and the DNA coiled on it constitute the nucleosome, serving as the basic structural unit of chromatin. The N-terminal tails of histone H2A, H2B, H3, and H4 may extensively modified post-translationally and serve as a hub receiving diverse regulatory signals from diet, lifestyle, and other environmental cues. With the technical improvements and scientific advancements, the list of modification types has expanded, encompassing not only classical acetylation, methylation, ubiquitination, and phosphorylation but also newly discovered ones, including lactylation and dopaminylation [33,34]. Among the classical modifications, acetylation, methylation, and ubiquitination primarily happen on lysine or arginine, while phosphorylation happens on serine or threonine. Distinct modification types may alter the chromatin structure and gene expression differentially. For example, histone acetylation generally correlates with transcriptional activation, whereas deacetylation tends to exert a transcriptionally repressive role. From the perspectives of their relevance in NAFLD, the classical histone modifications have been extensively investigated, and therefore, more details are summarized below.

### 2.1. Histone Methylation in NAFLD and its Therapeutic Implications

Histone methylation is catalyzed by histone methyltransferases (HMTs), and the histone demethylases (HDMs) remove the methylation marks on lysine or arginine residues. Notably, distinct lysine or arginine residuals require specific HMT and HDM enzymes, which also provide a strong specificity toward individual methyltransferase or demethylase, thereby reenforcing their distinct functional impacts.

Early evidence suggested histone H3 lysine 4 (H3K4) methyltransferase MLL2 could influence the metabolism via random ENU mutagenesis. A study revealed that a germline *Mll2* mutation led to insulin resistance and impaired glucose tolerance in mice [35]. In another study, HFD feeding led to the activation of ABL1 kinase, which phosphorylated PPARγ2 and enhanced the MLL4–PPARγ2 interaction. Consequently, overnutrition enhanced the recruitment of MLL4 to the promoter of PPARγ2-regulated steatosis target genes. This, in turn, increased the H3K4 methylation and transcriptional activation. Thus, the interaction between MLL4 and PPARγ2 proteins played a role in the development of fatty liver in HFD-fed mice [36,37] (Figure 3 and Figure 4). Moreover, the activation of hepatic stellate cells (HSCs), a key event in the transition from NAFLD to NASH, is also regulated by histone methylation. In activated HSCs, ASH1, another HMT for H3K4 methylation, directly bound to the regulatory genomic regions of alpha smooth muscle actin (α-SMA), collagen I, tissue inhibitor of metalloproteinase-1 (TIMP1), and transforming growth factor beta 1 (TGF-β1) to facilitate H3K4 methylation and transcriptional activation. Conversely, inhibiting ASH1 led to the downregulation of these fibrogenic gene expressions [38,39].

Protein arginine methyltransferase 5 (PRMT5) affects gene expression by methylating the arginine residues on histones, including H4R3, H3R8, and H2AR3. A previous study showed PRMT5 promoted the development of hepatic steatosis under a high-fat diet by facilitating the suppression of transcription regulators in mitochondrial biogenesis such as PPARα [40] (Figure 3A). PPARα functions as a pivotal transcription factor governing processes like fatty acid uptake, mitochondrial and peroxisomal fatty acid oxidation, and ketogenesis in the liver [41,42]. Mechanistically, PRMT5 was upregulated in the liver upon HFD. Conversely, the silencing or deletion of PRMT5 led to diminished AKT phosphorylation while increasing the expression of PPARα and PGC-1. This, in turn, elevated mitochondrial and peroxisomal fatty acid oxidation, demonstrating a propensity to slow down the development of fatty liver, although the mechanism with which PRMT5 regulates AKT phosphorylation remains unclear [40,43].

Methylations on K27 or 9 of histone H3 (H3K27 or H3K9 methylation) are typically linked to transcriptional repression. Polycomb repressive complex 2 (PRC2), a multicomponent protein complex, is the only methyltransferase for H3K27 and conserved from fungi to mammals. EZH2, the enzymatic subunit of PRC2, catalyzes the mono-, di-, and trimethylation of H3K27, which plays an important role in cell proliferation and differentiation. It has been reported that the EZH2 protein level was downregulated in the rat liver of a diet-induced NAFLD model and the fatty acid-induced insulin-resistant HepG2 model, although lacking a mechanism [44]. On the other hand, many studies have pointed toward out that EZH2/PRC2 activity or H3K27me3 may facilitate the disease progression to HCC via the repression of multiple tumor-suppressive microRNAs [45]. For example, it was reported that miR-200c was repressed by chromatin H3K27me3, and EZH2 depletion upregulated miR-200c and inhibited the growth of Huh7 in vitro and in vivo [46].

Unlike H3K27, there are multiple H3K9 methyltransferases, such as SUV39H2/KMT1B and the dimeric G9a, and they exhibit differential catalytic activities and target genes [47]. In methionine- and choline-deficient (MCD) diet-fed mice, SUV39H2 promoted hepatic steatosis by downregulating SIRT1, a NAD^+^-dependent histone deacetylase executing a protective role in the liver (more details in Section 2.4) [48]. Conversely, *Suv39h2* deletion alleviated diet-induced NASH in mice [49]. As H3K9 methylation is usually mechanistically associated with transcriptional repression, these effects may or may not be direct modulations and require further investigation. Nonetheless, the presented evidence underscores the role of histone methylation in the pathogenesis of NAFLD.

#### Potential Targets and Compounds Modulating Histone Methyltransferases in NAFLD

As NAFLD is a multifactorial chronic disease, treatment methods and approaches are still scarce despite the increased attention it has received. Since epigenetic modifications and NAFLD are closely related, altering histone modifications holds the potential to offer new opportunities in the treatment of NAFLD.

Targeting EZH2 has been intensively studied for its potential therapeutic implications in the treatment of NAFLD [50]. A fundamental event in the pathogenesis of hepatic fibrosis is the activation of quiescent HSC and their subsequent transformation into myofibroblasts. EZH2 has been reported to promote this transformation by suppressing the expression of PPARγ [51]. In addition, EZH2 was found to be inhibited by SIRT1, the protective NAD^+^-dependent histone deacetylase, and EZH2 inhibition is required for the protective effect of SIRT1 activation in myofibroblasts [52]. Furthermore, the inhibition of EZH2 decreased fibrogenic gene transcription in the TGF-β1-treated HSCs [53]. Indeed, DZNep, an HMT inhibitor, and GSK-503, a specific EZH2 inhibitor, prevent the progression of liver fibrosis in vivo by decreasing H3K27 methylation [44,50,53,54]. In addition, the herbal prescription Yang-Gan-Wan (YGW) and its active ingredients, rosmarinic acid (RA) and baicalin (BC), showed the potential to treat liver fibrosis by de-repressing *Ppar*γ in an epigenetic-dependent way, which suppressed the expression of EZH2 and reduced H3K27 di-methylation [55].

In addition, the expression of histone H3K9 methyltransferase G9a and the DNA methyltransferase DNMT1 was found to be upregulated in human cirrhotic liver and during mouse HSC activation. Using a dual chemical inhibitor of G9a and DNMT1 CM272, the authors showed that the inhibition of them simultaneously disrupted the profibrogenic metabolic reprogramming of HSC induced by TGF-β1 and inhibited liver fibrogenesis in vivo. Thus, the dual targeting of G9a and DNMT1 may provide a potential therapeutic approach for the treatment of liver fibrosis [56]. Targeting the complex epigenetic mechanisms involved in fibrogenesis with innovative molecules like CM272 may pave the way for better therapies.

### 2.2. Histone Demethylation in NAFLD and its Therapeutic Implications

In addition to HMTs, HDMs have been shown to be involved in NAFLD development. The histone demethylase Plant Homeodomain Finger 2 (PHF2) can erase the H3K9me2 mark. Mice with adenovirus overexpressing *Phf2* in the liver showed that increased levels of DAG and TG were protected from insulin resistance and inflammation [57] (Figure 3A). The reason was that the overexpression of *Phf2* could increase the level of SCD1, which catalyzes the desaturation of saturated fatty acids (SFA) to monounsaturated fatty acids (MuFA), and MuFA can prevent lipotoxicity. Consequently, PHF2 could prevent the progression to NASH with inflammation and fibrosis [57]. In addition, the lysine (K)-specific demethylase KDM7A belongs to the PHF2/PHF8 family of the Jumonji C (JmjC) domain-containing demethylase (JMJD demethylase) and has two identifiable domains: a PHD and a JmjC domain [58]. KDM7A overexpression could erase the H3K9me2 and H3K27me2 repressive markers on the DGAT2 promoter, thereby increasing the expression of DGAT2 and TG accumulation, which, finally, induced hepatic steatosis [59]. As SCD1 and DGAT2 enzymes are potential targets for the treatment of NAFLD and clinical trials are ongoing, PHF2 and KDM7A could provide potential therapeutic targets in treating NAFLD.

The KDM4 family is made up of four isoforms, KDM4A to D (also called JMJD2A to D). *KDM4A*, *B*, and *C* encode their respective proteins containing one JmjC, one JmjN, two PHD domains, and two Tudor domains. KDM4D is different from the other three isoforms, because it lacks both the PHD and Tudor domains [60,61]. KDM4D catalyzed the di-demethylation and tri-demethylation of H3K9, which stimulated TLR4 expression and triggered hepatic fibrogenesis by activating the NF-κB pathway. Meanwhile, KDM4D was significantly upregulated during HSC activation [60,62,63]. Unlike KDM4D, the other three isoforms were downregulated in HSC activation and facilitated the transcription of miR-29 together with SREBP2 to antagonize liver fibrosis [64]. In addition, KDM4B/JMJD2B could upregulate PPARγ2 and its target genes related to lipid droplet formation and fatty acid uptake by removing H3K9 methylation to promote hepatic steatosis [65]. A later study demonstrated that KDM4B plays a pivotal role in liver X receptor alpha (LXRα)-mediated lipogenesis [66], which provided another mechanism for KDM4B in hepatic steatosis (Figure 3A).

In addition to KDM4s, the role of the KDM3 subfamily in the liver has also been studied. KDM3A, KDM3B, and KDM3C (JMJD1A–C) are roughly 50% identical at the amino acid level and are all capable of removing dimethyl and monomethyl marks from H3K9 and H4R3 and nonhistone proteins, to a lesser extent. PPARγ expression was epigenetically regulated by KDM3A/JMJD1A during HSC activation (Figure 3C). KDM3A knockdown led to the elevated expression of fibrosis markers in HSCs and a mouse liver fibrosis model [67]. KDM3C/JMJD1C facilitated nutrient signaling to elevate the triglyceride levels in the liver and plasma by promoting the expression of lipogenesis genes [68]. Mechanistically, USF-1 recruited JMJD1C into multiple lipogenic genes in the fed state to demethylate H3K9me2 and increase chromatin accessibility (Figure 3A). JMJD1C was phosphorylated at T505 by the mTOR complex, which enhanced the direct interaction between USF-1 and JMJD1C and transduced the nutrient signal [68]. Therefore, a potential treatment approach for hepatosteatosis and IR is to target JMJD1C phosphorylation by mTOR, a critical lipogenic insulin signaling cascade [68]. KDM6B/JMJD3, a H3K27 demethylase, is another notable HDM in liver metabolism and NAFLD [69]. Under fasting conditions, JMJD3 and SIRT1 worked synergistically to activate fatty acid oxidation genes, such as *Fgf21*, *Cpt1a*, and *Mcad*. The liver-specific downregulation of JMJD3 reduced fatty acid oxidation and led to hepatic steatosis [70] (Figure 3B).

#### Potential Targets and Compounds Modulating Histone Demethylase in NAFLD

Small molecules that activate JMJD3 or promote the interaction of JMJD3 with SIRT1 specifically decreased the lipid levels, which may provide a therapeutic approach to treat obesity and hepatosteatosis [70]. Furthermore, fasting conditions also induced Fibroblast Growth Factor-21 (FGF21) signaling, which required JMJD3 to activate hepatic autophagy and lipid degradation through upregulating the global autophagy network genes [71] (Figure 4). This study also demonstrated that FGF21 administration to alleviate fatty liver in HFD mice was mediated by JMJD3 [71]. Thus, targeting the histone demethylase JMJD3 could be a potential treatment for NAFLD.

GSK2879552 is an LSD1 inhibitor that showed beneficial effect to inhibit FASN expression and ameliorate hepatic steatosis in mice. Mechanistically, the transcription factor Slug was upregulated in hepatocytes by insulin in fed state, which recruited the histone K3K9 demethylase LSD1 to *Fasn* promoter and promote FASN expression. So GSK2879552 blocked lipogenesis activated by Slug-LSD1 pathway and may be a useful therapeutic rationale for the treatment of NAFLD [72]. In addition, Gomisin N (GN) is a phytochemical from Schisandra chinensis, exhibiting hepatoprotective, anti-cancer, and anti-inflammatory properties [73]. In one study, the administration of GN was found to downregulate the expression of PPARγ2 and JMJD2B in the liver of HFD-induced obese mice [65], which may contribute to the alleviation of HFD-induced hepatic steatosis. GN was also found to reduce the tunicamycin-induced hepatic ER stress and TG accumulation in mice [74]. Thus, GN might be helpful for the treatment of NAFLD, although it may not work mainly through histone demethylation.

### 2.3. Histone Acetylation in NAFLD and Its Therapeutic Implications

Histone acetylation is a post-translational modification and mostly occurs at specific lysine residues in the N-terminal tails of the histone H3 and H4. Histone acetylation is always associated with chromatin opening and transcriptional activation. Histone acetylation is usually quite dynamic and jointly determined by histone acetyltransferases (HATs) and histone deacetylases (HDACs), which add or remove the acetyl group on particular histone lysine residues.

Histone acetylation may affect the expression of individual critical genes. In HepG2, H3 and H4 acetylation at fatty acid synthase gene *FASN* promoter was transiently increased upon insulin stimulation in a manner of cross-regulation with ChREBP, although the HAT was not identified [75]. In addition, it has been demonstrated that liver-specific knockdown of nuclear receptor subfamily 2, group F, member 6 (NR2F6) alleviated obesity-associated hepatosteatosis and MCD diet-induced NASH through downregulating CD36 expression in mouse models [76]. NR2F6 bound directly to *CD36* promoter in hepatocytes, recruited nuclear receptor coactivator 1 (SRC-1), a component of p300/CBP HAT complex, and promoted H3 acetylation on *CD36* promoter [76]. Interestingly, NR2F6 expression was increased in the livers of NAFLD patients and reduced by metformin treatment in obese mice [76]. Therefore, NR2F6 antagonists might offer a therapeutic approach for treating NAFLD through histone acetylation.

In addition, histone acetylation may be involved in the regulation of multiple genes/pathways simultaneously. Homozygous knock-in of a serine-to-alanine mutation at Ser196 (S196A) in LXRα to abolish the phosphorylation could affect the hepatic H3K27 acetylome and transcriptome during the progression of NAFLD. For example, the H3K27Ac at the *Ces1f* gene locus and the expression of *Ces1f* were high in the liver of LXRα-S196A mice comparing with WT mice when fed high-fat-high-cholesterol (HFHC) diet. *Ces1f* is a member of the carboxylesterase 1 family that controls hepatic lipid mobilization. Meanwhile, the H3K27Ac and expression of inflammation and fibrosis related genes including *Spp1* and *Col1e1* were reduced in S196A. So, LXRα-S196A could induce liver steatosis but prevent cholesterol accumulation, inflammation and fibrosis, thereby slowing the development from simple hepatic steatosis to NASH [77].

#### Potential Targets and Compounds Modulating Histone Acetylase in NAFLD

A few studies further suggested that histone acetylation can be a potential target for NAFLD. The active phosphorylated form of FTY720/fingolimod, a prodrug treating multiple sclerosis, could reduce *FASN* expression by histone acetylation alteration, inhibit ceramide production and hepatic steatosis in diet-induced NAFLD mice [78]. Tannic acid (TA), a HAT inhibitor, inhibited lipid accumulation in vivo and reduced the mRNA expression of genes associated with lipogenesis. Mechanistically, TA eliminated the occupancy of p300 on the sterol regulatory elements (SREs) in the promoters of *FASN* and ATP-citrate lyase (*ACLY*) genes, thereby decreasing acetylation of H3K9 and H3K36 [79].

The biguanide medicine metformin is the most popular anti-diabetic medication for the treatment of type 2 diabetes (T2D), which relieves hyperglycemia by reducing hepatic gluconeogenesis and improving insulin sensitivity [80,81,82]. It was reported that metformin activated AMPK, which directly phosphorylated and activated HAT1, promoted histone acetylation, and upregulated genes in mitochondria biogenesis [83]. Another study demonstrated that metformin promoted the phosphorylation of CBP at Ser436, which resulted in the dissociation of the CREB-CBP-TORC2 complex and downregulated the expression of the genes encoding gluconeogenic enzymes. Thus, metformin dramatically reduced the blood glucose level [84]. In addition, metformin increased hepatic protein levels of SIRT1 and GCN5 to inhibit hepatic gluconeogenesis [85].

### 2.4. Histone Deacetylation in NAFLD and Its Therapeutic Implications

The histone deacetylases (HDACs) family includes four distinct classes, namely class I, II, III and IV. HDAC1-3 and HDAC8 constitute Class I HDACs. Class II includes HDAC 4, 5, 6, 7, 9, and 10. On the other hand, class III HDAC enzymes, also known as sirtuins or silent information regulators (SIRT1-7), rely on NAD^+^ as a cofactor. Class IV HDAC exclusively consist of HDAC11 [86].

HDACs promote a dense chromatin structure and inhibit transcription by deacetylating lysine residues. In a diet-induced NAFLD and HCC mouse model, SREBP1 directly upregulated the expression of HDAC8, which worked with EZH2 concordantly through H4 deacetylation and H3K27 trimethylation to repress Wnt antagonists, thereby activating the Wnt pathway [87,88]. Snail1, a zinc-finger transcription factor, was reported to repress the expression of *FASN* through recruiting HDAC1/2 to deacetylate H3K9 and H3K27 at *FASN* promoter. In HFD-fed mice, Snail1 overexpression in the liver decreased the insulin-stimulated lipogenesis in hepatocytes and attenuated the fatty liver. Conversely, disruption of the insulin-snail1 pathway may lead to NAFLD (Figure 3B) [89]. HDAC6 was also reported to deacetylate the transcription factor FOXO1. S100 calcium binding protein A11 (S100A11) was upregulated in NAFLD liver, and then blocked the interaction between HDAC6 and FOXO1 to stimulate lipogenesis and liver steatosis [90]. Here the role of HDAC6 is beneficial along the line of NAFLD prevention.

SIRT1 is a NAD^+^ coenzyme-dependent histone deacetylase. Various studies have shown that SIRT1 can influence NAFLD through a variety of pathways. On one hand, SIRT1 is a unique upstream regulator of LKB1/AMPK sensing energy signaling [91]. Under fasting condition, the SREBP-1c acetylation level in mouse liver was consistently reduced and its interaction with SIRT1 was increased. The SREBP-1c-SIRT1 interaction was decreased after feeding, while SREBP-1c acetylation went up and better promoted lipogenesis [92]. In HFD model, liver-specific *Sirt1* knockout impaired PPARα/PGC-1α signaling and reduced fatty acid oxidation, thereby resulting in increased hepatic steatosis. This provided compelling evidence for a significant association between SIRT1 and hepatic fatty acid metabolism [93]. In diet-induced and genetically obese mice, pharmacological SIRT1 activators suppressed the hepatic lipid and cholesterol levels as well as liver steatosis [94]. In addition, SIRT1 could also inhibit NF-κB activity to reduce the inflammatory response, which is a powerful defender against pathologic conditions like fatty liver [95,96]. Thus, interventions stimulating SIRT1 activity could potentially offer therapeutic benefits for the management of hepatic diseases and metabolic syndrome associated with obesity [93].

Like SIRT1, SIRT6 has been implicated in the negative regulation of lipid metabolism and inflammation. Liver-specific Sirt6 knockout mice exhibited a tendency to increase hepatic steatosis, inflammation and insulin resistance under high-fat and high-fructose (HFHF) diet through upregulated BTB domain and CNC homolog 1 (Bach1), a nuclear repressor of Nrf2 [97]. In a similar study, liver-specific Sirt6 deletion led to fatty liver through reduced β-oxidation and enhanced glycolysis, lipogenesis and TG synthesis [98]. Mechanistically, SIRT1 interacts with FOXO3a and NRF1 on *SIRT6* promoter to positively regulate SIRT6 expression [98]. Indeed, multiple studies support the idea that SIRT6 promotes β-oxidation in a fasting state. In one study, SIRT6 regulated hepatic PPARα activity in vivo via deacetylation of the cofactor NCOA2 at K780 [99]. In parallel, SIRT6 could increase the activity of long-chain acyl-CoA synthase 5 (ACSL5) by deacetylating K98, K361 and K367 and promote fatty acid β-oxidation, thereby increasing the cellular lipid utilization and ultimately resisting the NAFLD process [100]. In addition, SIRT6 engaged in an interaction with acetyltransferase GCN5 and increased its activity, thereby suppressing hepatic gluconeogenesis. Therefore, hepatic SIRT6 activation may be therapeutically useful in the prevention of IR and NAFLD [101]. Another study demonstrated that SIRT6 overexpression in liver reduced steatosis, inflammation, and fibrosis caused by a HFHF diet, indicating the SIRT6 activator may be a promising therapeutic direction for treating NASH by reducing oxidative stress and inflammation [97]. Together, it is valuable to further explore the therapeutic agonists of SIRT1 and SIRT6 for the treatment of NAFLD.

#### Potential Targets and Compounds Modulating Histone Deacetylase in NAFLD

HDAC chemical inhibitors have been developed to treat cancer, and many of them have also been tested in mice models of NAFLD or obesity. The treatment of mice with valproic acid (VPA), a class I and II HDAC inhibitor, could decrease collagen deposition and HSC activation in the CCl_4_ model [102]. VPA is expected to have potential in preventing the further progression of liver fibrosis. Suberoylanilide hydroxamic acid (SAHA), another HDAC inhibitor, was shown to reduce liver fibrosis in rats through the suppression of TGF-β1 signaling [103]. In addition, sodium butyrate (NaB) increased H3K9Ac on the *PPARα* promoter, enhanced fatty acid oxidation, and inhibited the NF-κB inflammatory pathway, thereby alleviating NAFLD in the rat HFD model [104]. Thus, HDAC inhibitors may hold promise for the treatment of NAFLD.

Resveratrol (RSV), a natural polyphenol, not only exhibits anti-inflammatory and antioxidative characteristics but also activates SIRT1 [105]. RSV alleviated HFD-induced hepatic steatosis in mice liver and reduced lipid droplet accumulation in a SIRT1/ATF6-dependent manner [106]. In addition to the studies on mice models, there have been clinical studies demonstrating resveratrol has the potential to improve NAFLD in patients [107,108]. Patients receiving a daily dose of 300 mg resveratrol for 3 months had lower ALT and aspartate transaminase (AST) levels, higher lipid metabolism, and less inflammation [108]. On the other hand, it was also reported that patient treated with 3000 mg of resveratrol daily for 8 weeks did not show an improvement in their ALT and AST levels, hepatic steatosis, and insulin resistance [109]. Therefore, the dose and long-term effects of RSV require more study. Furthermore, SRT1720 is a specific SIRT1 activator that prevents diet-induced obesity and insulin resistance through enhancing fatty acid oxidation in the liver, muscle, and brown adipose tissue in mice [110]. SRT1720 has a significant protective effect against NAFLD in monosodium glutamate (MSG) mice. Treatment with SRT1720 reduced the expression of markers of oxidative stress, as well as inflammatory cytokines [111].

Interestingly, Olaparib, a poly (ADP-ribose) polymerase (PARP) inhibitor approved for cancer treatment, could increase hepatic SIRT1 activity/expression in mice [112] and showed the potential to treat fatty liver disorders. Although upregulating SIRT1 is only one aspect of the multifaceted mechanism of Olaparib, mice receiving Olaparib for 5 weeks showed significantly attenuated liver injury, inflammation, and fibrosis in a MCD diet model [112]. The monoterpenic phenol carvacrol (CVL), which exists in a variety of essential oils from the Labiatae family plants, also upregulates SIRT1. It has prospective hepatoprotective and neuroprotective effects [113]. Combined CVL and rosiglitazone treatment in HFD-fed mice improves the symptoms of diabetes mellitus, such as the reduction in the ALT/AST, plasma glucose, and insulin levels [114].

In addition to SIRT1 and SIRT6, SIRT3 is a mitochondrial sirtuin that has been investigated widely, which is essential for the maintenance of mitochondrial functions [115]. Protocatechuic acid (PCA), also known as 3,4-dihydroxybenzoic acid, is a natural phenolic compound in various food plants. Although it also has antioxidant and anti-inflammatory properties [116,117], PCA bound and upregulated the SIRT3 protein to prevent liver damage and steatosis in HFD-fed mice, likely by regulating the acetylation and degradation of Acyl-CoA synthetase family member 3 (ACSF3) and fatty acid metabolism [118].

### 2.5. Histone Ubiquitination and Phosphorylation in NAFLD

Histone ubiquitination is the process by which the ubiquitin molecule is specifically conjugated to histones on lysine residues in the presence of a family of enzymes, such as when activating, binding, and degrading ubiquitin. The ubiquitination of histones plays a role in changing the conformation of chromatin and recruiting and activating downstream readers or chromatin regulator proteins. For example, the overexpression of RNF20 effectively inhibited IL-6, TNFα, and VEGFA to prevent TGF-β-induced hepatic fibrosis via the ubiquitination of H2BK120 (H2BK120ub) [119]. Unlike acetylation and methylation, histone H3 serine 10 (H3S10) phosphorylation is a marker for mitotic chromatin and affected by the counteractions between kinases and phosphatases. Histone phosphorylation often cross-talks with other histone modifications and may regulate the chromatin status upon DNA damage and other stresses. For example, it was reported that ChREBP bound to ChoRE to upregulate the expression of *FASN* through the histone acetylation, methylation, and phosphorylation of histone H3 serine 10 (H3S10) [120], promoting hepatic steatosis. There are relatively scarce reports on histone ubiquitination and phosphorylation in NAFLD, and more studies may be in the queue.

## 3. Conclusions and Future Perspectives

NAFLD is a chronic and progressive hepatic disorder characterized by the increase of an excessive amount of fat in the liver. This accumulation of fat induces stress and damage to hepatocytes, resulting in inflammation and fibrosis. If left untreated, these pathological processes can ultimately lead to the development of liver injury, cirrhosis, hepatocellular carcinoma, and, ultimately, mortality [121]. Epigenetic mechanisms, specifically histone modifications here, are dynamic and can be reversibly regulated by a variety of external cues, such as nutrient signals. The role and mechanisms of histone modifications and the related enzymes in NAFLD have been investigated, and some of the interesting findings are summarized here (briefly listed in Table 1). Some of the useful compounds affecting these histone modifications are briefly listed in Table 2. Although they may not be directly useful in the treatment of patients, they provide specific tools for target validation in cell and animal models.

One of the noticeable features is that multiple histone modifications can be concordantly regulated and coordinately promote or inhibit specific gene networks. A few key cases are summarized in Figure 3 and may happen in different types of cells critical in NAFLD, such as hepatocytes and HSCs. Another feature worth mentioning is that many histone modifications are regulated by nutrient availability and status. Two cases are shown in Figure 4 to support this notion. More broadly, cofactors for histone modification enzymes include acetyl-CoA for acetyltransferases, S-adenosyl methionine for methyltransferases, and NAD^+^ for Class III HDACs. The concentrations of these cofactors are comprehensively affected by energy metabolism and other physiological/pathological conditions. In this way, histone modifications and other epigenetic marks may capture and reflect the integrated state of external inputs.

As histone modification plays an important role in the development of NAFLD and may be reversely regulated, there is increasing interest in the development of novel therapies focusing on modulating epigenetic variations [57]. Histone-modifying enzymes may provide targets for NAFLD therapy. However, NAFLD is a complex disease associated with multiple metabolic disorders, and its potential side effects still need to be considered. Therefore, although there is interest in the development of histone modification enzyme inhibitors as the treatment for NAFLD, more research needs to be done to examine the possible side effects, to discover target-selective inhibitors, and carefully assess their effectiveness in patients. Targeting the complex epigenetic mechanisms in NAFLD with dual-inhibitory molecules may also be tried. The prospective unexplored potential in histone modifications remains to be investigated and released in the future.

## Figures and Tables

**Figure 1 ijms-24-14653-f001:**
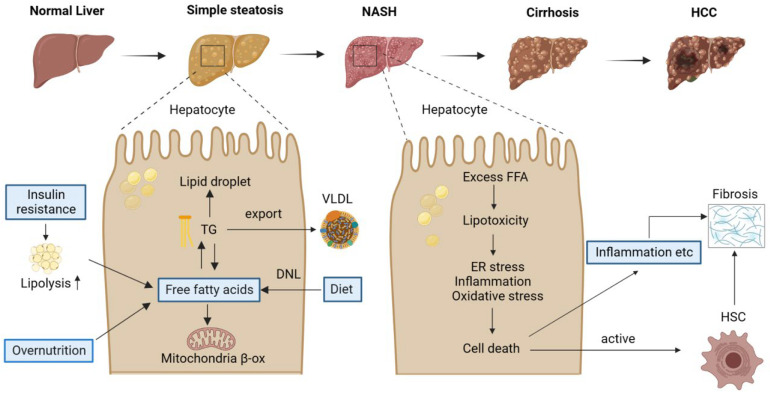
The pathological spectrum and pathogenesis of NAFLD. NAFLD is a progressive disease, including a broad spectrum of liver conditions, from simple steatosis to NASH, with the potential to progress to more severe stages such as cirrhosis and HCC. High-fat or -fructose diets could increase the free fatty acids in the liver. Free fatty acids have a key role in the development of NAFLD, and this proceeds three ways in the liver. (1) FFAs enter the mitochondria and are oxidized to produce energy and ketone bodies. (2) Esterified to TG and stored in lipid droplets. (3) Secreted and excreted as VLDL. NASH is characterized by steatosis, inflammation, and fibrosis. Cellular stresses due to lipotoxicity cause cell death and liver injury, which ultimately leads to hepatic fibrosis after the activation of HSCs. NASH, nonalcoholic steatohepatitis; HCC, hepatocellular carcinoma; TG, triglyceride; VLDL, very-low-density lipoprotein; β-ox, β-oxidization; DNL, de novo lipogenesis; FFA, free fatty acid; HSC, hepatic stellate cell.

**Figure 2 ijms-24-14653-f002:**
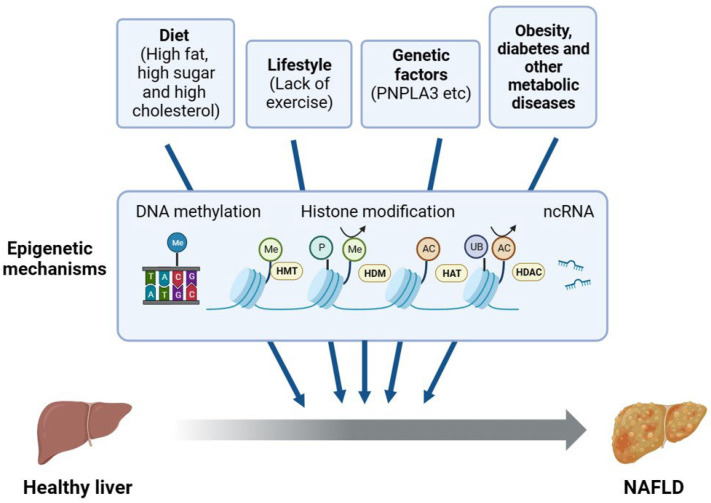
Epigenetic regulation of NAFLD. The factors influencing NAFLD include diet; lifestyle; and basal conditions such as metabolic syndrome, genetic variants, and epigenetic factors. In addition, non-epigenetic factors may influence NAFLD through epigenetic mechanisms, such as DNA methylation, histone modifications, and ncRNA. PNPLA3, patatin-like phospholipase domain-containing 3; HMT, histone methyltransferase; HDM, histone demethylase; HAT, histone acetyltransferase; HDAC, histone deacetylase; ncRNA, non-coding RNA.

**Figure 3 ijms-24-14653-f003:**
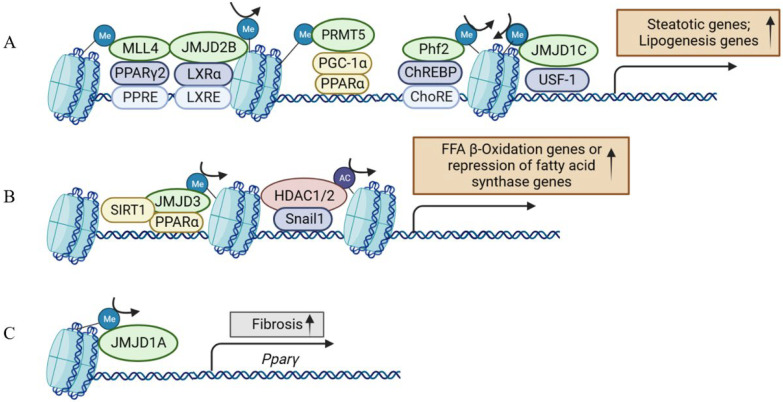
Histone-modifying enzymes and transcriptional regulators are concordantly involved in the multiple aspects of NAFLD development. (**A**) Histone-modifying enzymes involved in the process of hepatic steatosis. (**B**) Histone-modifying enzymes involved in the process of fatty acid β-oxidation or the repression of fatty acid synthase. (**C**) Histone-modifying enzymes involved in the process of hepatic fibrosis. The epigenetic regulation in (**A**,**B**) involves hepatocytes, and (**C**) involves hepatic stellate cells in addition to hepatocytes. MLL4, mixed lineage leukemia 4; PPARγ, peroxisome proliferator-activated receptor γ; PPRE, PPAR response element; JMJD2B, Jumonji domain-containing protein 2B; LXRα, liver X receptors α; LXRE, LXR response element; PRMT5, protein arginine methyltransferase 5; PGC-1α, peroxisome proliferator-activated receptor γ coactivator 1 alpha; PPARα, peroxisome proliferator-activated receptor α; Phf2, Plant Homeodomain Finger 2; ChREBP, carbohydrate-responsive element-binding protein; ChoRE, carbohydrate responsive element; JMJD1C, Jumonji domain-containing protein 1C; USF-1, upstream stimulatory factor 1; SIRT1, Sirtuin 1; HDAC1/2, histone deacetylases 1/2; JMJD1A, Jumonji domain-containing protein 1A.

**Figure 4 ijms-24-14653-f004:**
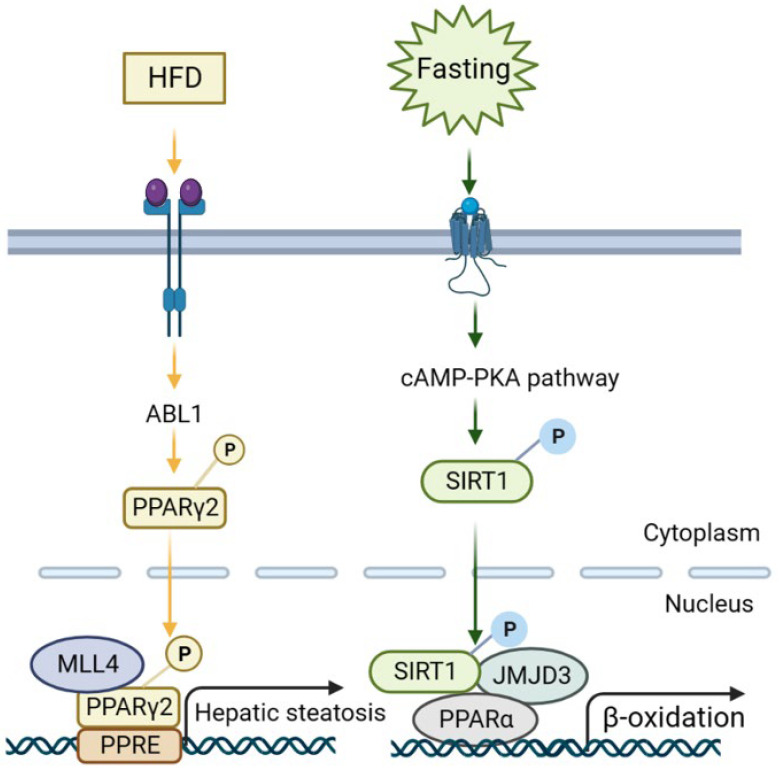
Representative signaling mechanisms of NAFLD regulation via histone-modifying enzymes. (1) Under HFD feeding, insulin and glucose signaling activates ABL1 kinase, which then phosphorylates PPARγ2, and the PPARγ2-MLL4 complex forms to promote hepatic steatotic target genes. (2) Fasting initiates cAMP/PKA signaling, resulting in the phosphorylation of SIRT1 and the formation of a JMJD3-SIRT1-PPARα complex in hepatocytes to increase the expression of its own gene and SIRT1-targeted β-oxidation via H3K27me3 demethylation.

**Table 1 ijms-24-14653-t001:** Summary of histone modifications and the corresponding mechanisms involved in NAFLD.

Epigenetic Regulators	Histone Modification	Effect	Reference
MLL4	H3K4me1	Resulting in the induction of the steatosis target genes of PPARγ2; UBE3A degrades MLL4 to suppress steatosis	[36,37]
ASH1	H3K4me3	ASH1 inhibition results in the downregulation of fibrogenic gene expression	[38,39]
PRMT5	Symmetric dimethylation of H4R3, H3R8, and H2AR3	PRMT5 inhibition increases hepatic triglyceride levels	[40,43]
SUV39H2	H3K9me3	Facilitating hepatic steatosis by repressing SIRT1 transcription	[48,49]
PHF2	H3K9me2 demethylation	Preventing lipotoxity via upregulating SCD1	[57]
KDM7A	H3K9me2 and H3K27me2 demethylation	Inducing hepatic steatosis by increasing the expression of DGAT2	[59]
KDM4D	H3K9me demethylation	Triggering hepatic fibrogenesis by activating NF-κB signaling pathways; Promoting HCC progression	[60,62,63]
KDM4A, KDM4B, KDM4C	H3K9me demethylation	SREBP2-KDM4 pathway leads to the maintenance of HSC quiescence by promoting miR-29 transcription	[64]
KDM4B	H3K9 demethylation	Upregulating PPARγ and target genes related to fatty acid uptake	[65,66]
KDM3A	H3K9 demethylation	Downregulating the KDM3A increases liver fibrosis through modulating the expression of PPARγ	[67]
KDM3C	H3K9 demethylation	Elevating TG levels in the liver and plasma by upregulating lipogenesis genes	[68]
KDM6B	H3K27me3 demethylation	Liver-specific downregulation of KDM6B reduces fatty acid oxidation and leads to NAFLD; KDM6B inhibition enhances HSCs activation	[69,70]
HDAC8	H4 deacetylation	Suppressing Wnt antagonists together with EZH2; AHR-HDAC8 axis promotes HCC	[87,88]
HDAC1/2	H3K9 and H3K27 deacetylation	Recruited to fatty acid synthase promoter by snail1 and decrease insulin-stimulated lipogenesis	[89]

**Table 2 ijms-24-14653-t002:** Summary of histone modification-modulating compounds tested in NAFLD models.

Chemical Compounds	Target	Effect	Reference
GSK-503	EZH2	Preventing the progression of liver fibrosis in mice	[53]
DZNep			[54]
Yang-Gan-Wan (YGW)	EZH2	Inhibiting liver fibrosis by de-repressing *Pparγ* in mice	[55]
CM272	G9a and DNMT1	Inhibiting liver fibrogenesis in mice and human	[56]
GSK2879552	LSD1	Downregulating LSD1-mediated Slug stimulation of FAS expression in mice	[72]
Gomisin N	JMJD2B	Alleviating HFD-induced hepatic steatosis and reducing hepatic ER stress in mice	[65,74]
Tannic acid	HAT	Inhibiting lipid accumulation in mice	[79]
Metformin	HATs (p300/CBP/ GCN5)	Reducing the blood glucose level and inhibiting hepatic gluconeogenesis in mice	[83,84,85]
Valproic acid	HDAC	Inhibiting HSC activation in mice	[102]
SAHA	HDAC	Reducing liver fibrosis in rats	[103]
NaB	HDAC	Enhancing fatty acid oxidation and inhibiting NF-κB inflammatory pathway in rats	[104]
Resveratrol	SIRT1	Alleviating hepatic steatosis in mice; reducing ALT/AST and improving insulin resistance in human with NAFLD	[106,107,108]
SRT1720	SIRT1	Decreasing the expressions of marker genes for oxidative stress and inflammatory cytokines and ameliorating fatty liver in mice	[110,111]
Olaparib	SIRT1	Decreasing hepatic triglyceride accumulation, inflammation and fibrosis in NASH mice	[112]
Carvacrol	SIRT1	Decreasing hepatic marker enzymes (ALT/AST) activities in combination with rosiglitazone in mice	[114]
Protocatechuic acid	SIRT3	Preventing liver damage and steatosis in mice and rats	[118]

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
