# Peer review of "Histone Modifications in NAFLD: Mechanisms and Potential Therapy"

_ijms, 2023, doi:10.3390/ijms241914653_

Round 1

Reviewer 1 Report

The submitted review tackles a very important issue, i.e. the role of epigenetic regulation in the development of NAFLD, and the possible therapeutic approaches that could be used to prevent or treat this widely spread disease, which could evolve into cirrhosis or HCC. The authors focused on the most studied modification, i.e. methylation, acetylation and ubiquitination, reviewing in depth the enzymes involved in the various processes. Although the general structure of the review is fine, there are some issues that must be addressed. First of all there must be a profound revision of the language, since there are several errors also in the sentence construction that prevent the reader to correctly understand the meaning. Secondly, the paper is a review but a lot of the references are themselves reviews and not the original papers. Although some reviews are acceptable, most of the references should be those reporting the findings on which the authors are deriving their considerations.

In particular:

lines 97-9: “PNPLA3 protein is involved in the lipolysis of lipid droplets in hepato-97 cytes, and this I148M variant interferes with normal turnover of PNPLA3 and therefore 98 exhibited a robust correlation with hepatic fat content”  Please define the type of effect, since the sentence is not clear

lines 111-115 “For example, the cytosine methylation (5mC) of mitochondrial DNA (mtDNA) in hepatocytes contributes to the disease progression from simple steatosis to NASH [23]. In another study, miR-20b was shown to specifically target PPARα to regulate hepatic lipid metabolism and 114 NAFLD [24].” As above, please expand the sentences in order to make more understandable the meaning of the findings.

Lines 151-158  please rephrase the sentences in order to make them clearer

Lines 183-186 please provide more detailed information.

Lines 190 and 204 SIRT1 is mentioned but its role is defined much later in the review. It would be better to briefly explain it here as well to make it easier for the reader

Lines 285-288 “In addition, it has been demonstrated that liver-specific knockdown of nuclear receptor subfamily 2, group F, member 6 (NR2F6) alleviated obesity-associated hepatosteatosis and MCD diet-induced NASH through upregulating CD36 expression in mouse models [54].”  Please check the reference, since CD36 should be downregulated.

Lines 295-300 please explain better the reference, since the authors screened for the H3K27Ac but the correlation with the mutated form of LXRa needs to be clarified.

see full comment

Author Response

Reviewer 1#

  1. First of all there must be a profound revision of the language, since there are several errors also in the sentence construction that prevent the reader to correctly understand the meaning.

Respond: We thank the reviewer for pointing this out. We have carefully revised the manuscript and all the revisions are marked in the a tracking version (also uploaded).

  1. Secondly, the paper is a review but a lot of the references are themselves reviews and not the original papers,Although some reviews are acceptable, most of the references should be those reporting the findings on which the authors are deriving their considerations.

Respond: We appreciate this comment and checked the references carefully. In this revised version, 12 reviews were removed from the reference and the relevant original research papers are cited now.

  1. (1) Lines 97-9: “PNPLA3 protein is involved in the lipolysis of lipid droplets in hepatocytes, and this I148M variant interferes with normal turnover of PNPLA3 and therefore exhibited a robust correlation with hepatic fat content” Please define the type of effect, since the sentence is not clear

Respond: We have revised the sentence in this version of manuscript. The details as follow.

“PNPLA3 is involved in the lipolysis of lipid droplets in hepatocytes. The I148M variant escapes proteasome degradation, accumulates on lipid droplets, and blocks the function of adipose triglyceride lipase and lipolysis. PNPLA3-I148M exhibited a robust correlation with increased hepatic TG accumulation and hepatic inflammation in human genome-wide association studies. PNPLA3-I148M expression in liver resulted in an increase in liver triglyceride content in a transgenic mouse model.’’ (Lines 99-106).

(2) Lines 111-115 “For example, the cytosine methylation (5mC) of mitochondrial DNA (mtDNA) in hepatocytes contributes to the disease progression from simple steatosis to NASH [23]. In another study, miR-20b was shown to specifically target PPARα to regulate hepatic lipid metabolism and 114 NAFLD [24].” As above, please expand the sentences in order to make more understandable the meaning of the findings.

Respond: We have corrected the sentences as follows.

“For example, the cytosine methylation (5mC) of mitochondrial DNA (mtDNA) had been found in the liver biopsies of patients with NAFLD, and patients with NASH had higher level of DNA methyltransferase 1 [29]. In another study, AAV- miR-20b administration induced hepatic steatosis and reduced FA oxidation in HFD-fed mice, possibly by decreasing the level of PPARα.’’ (Liness 116-120)

(3) Lines 151-158 please rephrase the sentences in order to make them clearer

Respond: We have rephrased this part to make the sentences clearer.

“In another study, the HFD feeding led to the activation of ABL1 kinase, which phosphorylated PPARγ2 and enhanced the MLL4 -PPARγ2 interaction. Consequently, overnutrition enhanced the recruitment of MLL4 to the promoter of PPARγ2-regulated steatosis target genes. This, in turn, increased the H3K4 methylation and transcriptional activation. Thus, the interaction between MLL4 and PPARγ2 proteins played a role in the development of fatty liver in HFD-fed mice’’ (Lines 160-165)

(4) Lines 183-186 please provide more detailed information.

Respond: We thank the reviewer for this helpful suggestion. More details were added in the revised manuscript as below.

“On the other hand, many studies pointed toward the direction that EZH2/PRC2 activity or H3K27me3 may facilitate the disease progression to HCC via repression of multiple tumor suppressive microRNAs. For example, it was reported that miR-200c was repressed by chromatin H3K27me3, and EZH2 depletion upregulated miR-200c and inhibited the growth of Huh7 in vitro and in vivo.” (Lines 192-196)

(5) Lines 190 and 204 SIRT1 is mentioned but its role is defined much later in the review. It would be better to briefly explain it here as well to make it easier for the reader

Respond: We thank the reviewer for this helpful suggestion. The manuscript is revised accordingly. Brief explanations on SIRT1 were added in lines 200-201 and lines 215-216.

(6) Lines 285-288 “In addition, it has been demonstrated that liver-specific knockdown of nuclear receptor subfamily 2, group F, member 6 (NR2F6) alleviated obesity-associated hepatosteatosis and MCD diet-induced NASH through upregulating CD36 expression in mouse models [54].” Please check the reference, since CD36 should be downregulated.

Respond: We were sorry for this mistake and thank the reviewer for pointing this out. We made the correction in this revision.

(7) Lines 295-300 please explain better the reference, since the authors screened for the H3K27Ac but the correlation with the mutated form of LXRa needs to be clarified.

Respond: We thank the reviewer for this helpful suggestion. The manuscript is revised as below.

“In addition, histone acetylation may be involved in regulation of multiple genes/pathways simultaneously. Homozygous knock-in of a serine-to-alanine mutation at Ser196 (S196A) in LXRα to abolish the phosphorylation could affect the hepatic H3K27 acetylome and transcriptome during the progression of NAFLD. For example, the H3K27Ac at the Ces1f gene locus and the expression of Ces1f were high in the liver of LXRα-S196A mice comparing with WT mice when fed high-fat-high-cholesterol (HFHC) diet. Ces1f is a member of the carboxylesterase 1 family that controls hepatic lipid mobilization. Meanwhile, the H3K27Ac and expression of inflammation and fibrosis related genes including Spp1 and Col1e1 were reduced in S196A. So, LXRα-S196A could induce liver steatosis but prevent cholesterol accumulation, inflammation, and fibrosis, thereby slowing the development from simple hepatic steatosis to NASH.” (Lines 324-334)

Reviewer 2 Report

 Dear Authors,

although there are  already some  reviews dealing with epigenetics in NAFLD, the focus on  histone modifcations  is of high relevance.  The  review is well written and I have only some concerns:

1)  Table 1 is very  helpful , well structured, and very interesting.  If possible, the authors should also refer to the primary and original references in the text and not to other reviews .

2)  In my oppinion the information of  figure 2 is low.

3) The therapeutic aspects , which are mentioned in the  title  and in  some subtitles, should be highlighted  and it should be mentioned, what kind of therapy would be conceivable.

 Dear Editor , the review is well written and I have only minor comments concerning the  references, the therapeutic aspects, and the figure 2.   Therefore I recommend  to accept the  manuscript  after minor  revisions.  With kind regards

Margarete odenthal

Author Response

Reviewer 2#

  1. Table 1 is very helpful, well structured, and very interesting. If possible, the authors should also refer to the primary and original references in the text and not to other reviews.

Respond: We thank the reviewer for the constructive suggestion. We checked carefully and added in more original studies in the reference of table 1. The reference of the revised manuscript was also revised accordingly.

  1. In my oppinion the information of figure 2 is low.

Respond: We thank the reviewer for pointing out this caveat. We revised figure 2 to add in more information by listing the three main categories of epigenetic mechanisms in details.  

  1. The therapeutic aspects, which are mentioned in the title and in some subtitles, should be highlighted and it should be mentioned, what kind of therapy would be conce

Respond: We thank the reviewer for this constructive suggestion. We now added subtitles “Potential therapeutic implications of histone modification in NAFLD” in each sections of histone medication and summarized the compound information in Table 2. In this way, the therapeutic aspects are summarized and emphasized better.

Round 2

Reviewer 1 Report

No further comments